# A Robustness Test for Estimating Total Effects with Covariate Adjustment

**Zehao Su**[1]                    **Leonard Henckel**[2]

[1]Section of Biostatistics, Department of Public Health, University of Copenhagen
[2]Department of Mathematical Sciences, University of Copenhagen

## Abstract

Suppose we want to estimate a total effect with covariate adjustment in a linear structural equation model. We have a causal graph to decide what covariates to adjust for, but are uncertain about the graph. Here, we propose a testing procedure, that exploits the fact that there are multiple valid adjustment sets for the target total effect in the causal graph, to perform a robustness check on the graph. If the test rejects, it is a strong indication that we should not rely on the graph. We discuss what mistakes in the graph our testing procedure can detect and which ones it cannot and develop two strategies on how to select a list of valid adjustment sets for the procedure. We also connect our result to the related econometrics literature on coefficient stability tests.

## 1 INTRODUCTION

Suppose we are interested in estimating the total (causal) effect of a treatment $X$ on an outcome $Y$ from observational data. One popular approach to estimate such an effect is covariate adjustment, also known as adjusting for confounding. Deciding which covariates to adjust for is a difficult problem, but it can be answered precisely if we have knowledge of the underlying causal structure in the form of a graph [Pearl, 2009]. In particular, the class of covariate sets we may adjust for has been fully graphically characterised [Shpitser et al., 2010, Perković et al., 2018]. We refer to sets in this class as valid adjustment sets.

In some cases there is more than one valid adjustment set, which raises the question how we can exploit this. One approach is to try and select from the available valid adjustment sets the one that provides the most statistically efficient estimator [Kuroki and Miyakawa, 2003, Rotnitzky and Smucler, 2020, Witte et al., 2020, Henckel et al., 2022].

Another natural approach is to use multiple valid adjustment sets and test whether they all in fact lead to estimators of the same quantity. Such a test would be a simple and targeted robustness check on the causal graph we are relying on. Here we say targeted in the sense that the test would only detect mistakes in the graph that are relevant to our goal of estimating the total effect of interest, which is easier than checking whether the entire graph is correct.

In the econometrics literature, it is already common practice to estimate the total effect with multiple estimators and then check whether the estimates differ by a large margin [e.g. Dikova et al., 2019, Yigezu and El-Shater, 2021, Schlegel et al., 2021]. This approach is often called testing for coefficient stability [Walter and Tiemeier, 2009] or simply called a robustness check [Lu and White, 2014].

Practitioners often verify coefficient stability in a heuristic manner, but there is also a theoretical literature on the topic, especially for instrumental variable estimators [Frank et al., 2013, Oster, 2019].

For covariate adjustment estimators, Lu and White [2014] have proposed a formal test for coefficient stability. Their framework is not based on graphical models and therefore it is harder to decide which adjustment sets to use for their test. They propose to fix what they call a core of covariates and then create additional adjustment sets by adding what they call non-core covariates to the core. Here, the status of being core or non-core depends on certain conditional independences. In the graphical framework it becomes clear that this approach is too restrictive as, for example, two valid adjustment sets may be disjoint. As a result their approach may consider too few sets which leads to a loss of power.

There exists a more general literature on validation tests for structural equation models. This literature, however, has focused on tests that either validate the entire model [Bollen, 1989, Bollen and Ting, 1993, Spirtes et al., 2000] or rely on instrumental variables [Kirby and Bollen, 2009]. Another related literature focuses on identifying valid adjustment sets by relying on an auxiliary variable, typically called

*Accepted for the 38th Conference on Uncertainty in Artificial Intelligence* (UAI 2022).

an anchor, whose causal relationship to the treatment has to be known from domain knowledge [Entner et al., 2013, Gultchin et al., 2020, Shah et al., 2021, Cheng et al., 2022].

In this paper, we adopt the framework of a linear structural equation model compatible with an unknown directed acyclic graph (DAG). We propose a targeted robustness test that given a pair $(X, Y)$ and a candidate DAG $\mathcal{G}$ tests whether the valid adjustment sets with respect to $(X, Y)$ in $\mathcal{G}$ lead to estimates of the same quantity.

We first discuss, which mistakes in the candidate graph our robustness test has power for and how this depends on the valid adjustment sets we use for the test. We then propose a simple $\chi^2$-test, similar to the one proposed by Lu and White [2014], although it differs in that we do not require a fixed core of covariates. We show that in general the joint asymptotic covariance matrix of the estimators we wish to compare is degenerate and that we need to know its rank for the test. This problem was acknowledged but not addressed by Lu and White [2014].

In response to the problem of the degenerate asymptotic covariance matrix, we propose two strategies. The first is to estimate the rank. This is a difficult statistical problem and may be unstable, especially in small samples. The upside of this approach, however, is that it allows us to use all valid adjustment sets for our test, which maximises its power.

The second strategy carefully selects a subset of the valid adjustment sets in a way that ensures the following two properties are likely to hold. First, the asymptotic covariance matrix is not degenerate. Second, we do not lose power completely against mistakes in the graph we had power for when using all valid adjustment sets. These two strategies represent different trade-offs between the stability of our testing procedure and its power to detect mistakes in the candidate graph.

Finally, we investigate with a simulation study how well our testing procedure controls the type-I error rate and how much power it has in finite samples. We do so for both of the strategies we propose, in order to compare and contrast their respective advantages and disadvantages. We also illustrate our testing procedure on a real data problem. All proofs are given in the supplementary materials. An implementation of our testing procedure and the code for our simulation study are made available at https://github.com/zehaosu/RoCA.

## 2 PRELIMINARIES

We consider a linear structural equation model compatible with a DAG, where nodes represent random variables and edges represent direct effects. We now provide the most important definitions. The remaining definitions are provided in Section A of the supplementary materials.

*Linear structural equation models.* Let $\mathcal{G} = (\mathbf{V}, \mathbf{E})$ be a DAG. Then $\mathbf{V} = (V_1, \ldots, V_p)$ follows a *linear structural equation model* compatible with $\mathcal{G}$ if for all $i = 1, \ldots, p$

$$V_i \leftarrow \sum_{V_j \in \mathrm{pa}(V_i, \mathcal{G})} \alpha_{ij} V_j + \epsilon_i,$$

with edge coefficients $\alpha_{ij}$ and jointly independent errors $\epsilon_i$ with zero mean and finite variance. We do not assume that the errors are normally distributed.

*Total effects.* Consider a pair $(X, Y)$ of random variables. The *total effect* of $X$ on $Y$ is the partial derivative of the expectation $\mathrm{E}(Y \mid do(X = x))$ with respect to $x$. This is the instantaneous change of the average of $Y$ in the world where $X$ is set to $x$ [Pearl, 2009]. In a linear structural equation model, the partial derivative is a constant slope that does not depend on $x$. As a result, the total effect is simply a number $\tau_{yx}$.

*Causal and forbidden nodes.* Consider two nodes $X$ and $Y$ in a DAG $\mathcal{G} = (\mathbf{V}, \mathbf{E})$. The *causal nodes* relative to $(X, Y)$ in $\mathcal{G}$, denoted $\mathrm{cn}(X, Y, \mathcal{G})$, are all nodes on directed paths from $X$ to $Y$, excluding $X$. The descendants of $X$ in $\mathcal{G}$, denoted $\mathrm{de}(X, \mathcal{G})$, are all nodes $V$ such that there exists a directed path from $X$ to $V$ in $\mathcal{G}$. The *forbidden nodes* relative to $(X, Y)$ in $\mathcal{G}$, denoted $\mathrm{forb}(X, Y, \mathcal{G})$, are all nodes that are descendants of causal nodes, including $X$. The *non-forbidden nodes* relative to $(X, Y)$ in $\mathcal{G}$, denoted $\mathrm{nonforb}(X, Y, \mathcal{G})$, are the nodes in $\mathbf{V} \setminus \mathrm{forb}(X, Y, \mathcal{G})$.

*Notation for regression coefficients.* Consider random variables $X$ and $Y$, random vectors $\mathbf{Z}_1, \ldots, \mathbf{Z}_k$ and the collection of adjustment sets $\mathcal{Z} = \{\mathbf{Z}_1, \ldots, \mathbf{Z}_k\}$. Let $\beta_{yx.\mathbf{z}_i}$ indicate the population level regression coefficient of $X$ in the ordinary least squares regression of $Y$ on $X$ and $\mathbf{Z}_i$. Let $\hat{\beta}_{yx.\mathbf{z}_i}$ denote the corresponding estimator. Let $\boldsymbol{\beta}_{yx.\mathcal{Z}}$ denote the stacked population regression coefficients $(\beta_{yx.\mathbf{z}_1}, \ldots, \beta_{yx.\mathbf{z}_k})^\top$ and $\hat{\boldsymbol{\beta}}_{yx.\mathcal{Z}}$ the corresponding estimator. Finally, let $\delta_{yz_i} = Y - \beta_{yz_i} Z$ be the population level residuals for the ordinary least squares regression of $Y$ on $Z_i$ and $r_{yz_i}$ be the corresponding vector of sample residuals.

*Valid adjustment sets.* Consider nodes $X$ and $Y$ in a DAG $\mathcal{G} = (\mathbf{V}, \mathbf{E})$. A node set $\mathbf{Z}$ is a *valid adjustment set* relative to $(X, Y)$ in $\mathcal{G}$ if for all linear structural equation models compatible with $\mathcal{G}$, $\beta_{yx.\mathbf{z}} = \tau_{yx}$. We say a valid adjustment set $\mathbf{Z} = \{Z_1, \ldots, Z_k\}$ is *minimal* if for all $i \in \{1, \ldots, k\}$, $\mathbf{Z} \setminus Z_i$ is not a valid adjustment set. The class of valid adjustment sets has been fully characterised as follows.

*Adjustment criterion.* [Shpitser et al., 2010, Perković et al., 2018] A (possibly empty) set $\mathbf{Z}$ is a valid adjustment set relative to $(X, Y)$ in $\mathcal{G}$ if and only if

1. $\mathbf{Z}$ contains no node in $\mathrm{forb}(X, Y, \mathcal{G})$, and

2. $\mathbf{Z}$ blocks all paths between $X$ and $Y$ in $\mathcal{G}$ that are not directed from $X$ to $Y$.

*d-separation.* Consider three disjoint node sets $\mathbf{X}, \mathbf{Y}$ and $\mathbf{Z}$ in a DAG $\mathcal{G} = (\mathbf{V}, \mathbf{E})$, such that $\mathbf{V}$ follows a linear structural equation model compatible with $\mathcal{G}$. We can read off from $\mathcal{G}$ whether $\mathbf{X}$ is independent of $\mathbf{Y}$ given $\mathbf{Z}$ with a graphical criterion called *d-separation* [Pearl, 2009] which we define formally in the supplementary materials. We use the notation $\mathbf{X} \perp_{\mathcal{G}} \mathbf{Y} \mid \mathbf{Z}$ to denote that $\mathbf{X}$ is d-separated from $\mathbf{Y}$ given $\mathbf{Z}$ in $\mathcal{G}$.

## 3 A TARGETED ROBUSTNESS TEST FOR COVARIATE ADJUSTMENT

### 3.1 THE NULL HYPOTHESIS AND ITS PROPERTIES

Suppose we wish to estimate the total effect of a treatment $X$ on a response variable $Y$. Let $\mathcal{G}_0$ denote the unknown true underlying casual graph and suppose we have a candidate causal graph $\mathcal{G}$ that describes our understanding of the underlying causal structure but that we are not certain about. We would like to check whether the candidate graph is plausible, so we can rely on it to estimate $\tau_{yx}$ with some confidence. In order to do so, we use $\mathcal{G}$ to identify a collection of valid adjustment sets $\mathcal{Z} = \{\mathbf{Z}_1, \ldots, \mathbf{Z}_k\}$ with respect to $(X, Y)$ in $\mathcal{G}$. If $\mathcal{G}$ is correct each of these sets corresponds to a consistent estimator of $\tau_{yx}$, i.e.,

$$\tau_{yx} = \beta_{yx.\mathbf{z}_1} = \beta_{yx.\mathbf{z}_2} = \cdots = \beta_{yx.\mathbf{z}_k}. \tag{1}$$

If $\mathbf{Z}$ consists of more than one set, then equation (1) imposes an over-identifying constraint on the total effect $\tau_{yx}$. We use this constraint to test the plausibility of the candidate graph $\mathcal{G}$. The more valid adjustment sets $\mathbf{Z}_i$ we use, the more mistakes in $\mathcal{G}$ the test can detect.

It is generally not possible to directly test the constraint from equation (1) with observational data because we do not know the true total effect $\tau_{yx}$. It is, however, possible to test the relaxed null hypothesis

$$H_0 : \beta_{yx.\mathbf{z}_1} = \beta_{yx.\mathbf{z}_2} = \cdots = \beta_{yx.\mathbf{z}_k}.$$

Let $H_0^*$ denote the null hypothesis associated with equation (1). As $H_0$ holds whenever $H_0^*$ does, it follows that any test with type-I error rate control for testing $H_0$ also has type-I error rate control for testing $H_0^*$. In addition, any rejection of $H_0$ implies a rejection of $H_0^*$ and as a result of the candidate graph $\mathcal{G}$. It is therefore reasonable to test $H_0$ as a proxy for $H_0^*$.

There is an even more restrictive null hypothesis $H_0^{**} : \mathcal{G} = \mathcal{G}_0$. However, as we are interested in estimating one specific total effect, it is not necessary to validate the entire candidate graph $\mathcal{G}$, and the distinction between $H_0^*$ and $H_0^{**}$ is irrelevant for the purposes of this paper. There are, however, cases where $H_0$ holds but $H_0^*$ does not and in these cases any test for $H_0$ will have no power to reject

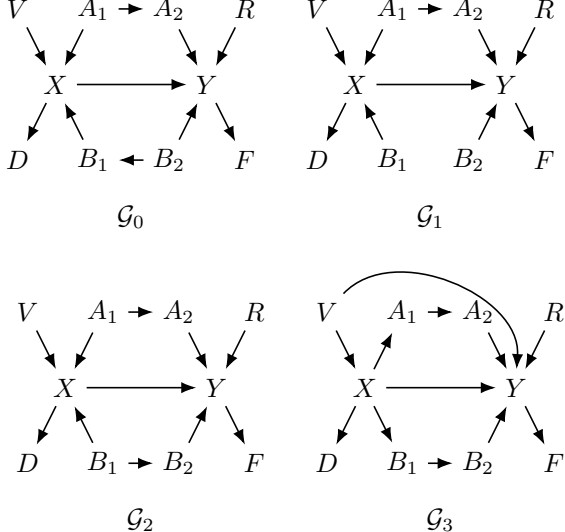

Figure 1: Graphs used in Examples 2, 4 and 9.

$H_0^*$. This occurs whenever $H_0$ holds but for all $\mathbf{Z}_i \in \mathcal{Z}$, $\beta_{yx.\mathbf{z}_i} \neq \tau_{yx}$. Whether this is the case depends on the choice of candidate sets $\mathcal{Z}$ and is more likely if $\mathcal{Z}$ contains few sets. In particular, this is impossible if $\mathcal{Z}$ contains even a single valid adjustment set from the true graph $\mathcal{G}_0$. In response, it is natural to use all available valid adjustment sets relative to $(X, Y)$ in $\mathcal{G}$ to maximise the number of sets in $\mathcal{Z}$. However, even for a moderately sized $\mathcal{G}$ the number of valid adjustment sets relative to the pair $(X, Y)$ can be very large, e.g., there are 72 in the graph $\mathcal{G}_0$ and 96 in the graph $\mathcal{G}_1$ from Figure 1. This raises the question whether it is possible to select the collection $\mathcal{Z}$ in a way that minimises the risk of having no power for $H_0^*$, while simultaneously limiting its size.

We now provide a necessary condition on $\mathcal{Z}$ under which the problematic case that $H_0$ holds but $H_0^*$ does not is rare, and as a result testing $H_0$ is a good proxy for testing $H_0^*$.

**Theorem 1.** *Consider nodes $X$ and $Y$ in a DAG $\mathcal{G}_0 = (\mathbf{V}, \mathbf{E})$ such that $Y \in \mathrm{de}(X, \mathcal{G}_0)$. Let $\mathcal{Z} = \{\mathbf{Z}_1, \ldots, \mathbf{Z}_k\}$ be a collection of node sets in $\mathcal{G}_0$. Suppose there exists a $\mathbf{Z}_i$, such that $\mathrm{forb}(X, Y, \mathcal{G}_0) \cap \mathbf{Z}_i = \emptyset$ and $\mathbf{Z}_i$ is not a valid adjustment set relative to $(X, Y)$ in $\mathcal{G}_0$. Further, suppose that $(\mathbf{V} \setminus \mathrm{forb}(X, Y, \mathcal{G}_0)) \subseteq \bigcup_{j=1}^{k} \mathbf{Z}_j$. If we sample the edge coefficients and error variances for a linear structural equation model compatible with $\mathcal{G}_0$ from a distribution $P$ such that $P$ is absolutely continuous with respect to the Lebesgue measure, then $P$-almost surely there exists a $\mathbf{Z}_j$ such that $\beta_{yx.\mathbf{z}_i} \neq \beta_{yx.\mathbf{z}_j}$.*

Verifying that Theorem 1 holds requires knowledge of the true DAG $\mathcal{G}_0$, which we do not have. Nonetheless, it gives two important but also intuitive insights on how to select $\mathcal{Z}$. First, the sets in $\mathcal{Z}$ should cover as many nodes as possible, i.e., ideally all non-forbidden nodes in the candidate graph

$\mathcal{G}$. This maximises the chances that $(\mathbf{V} \setminus \text{forb}(X, Y, \mathcal{G}_0)) \subseteq \bigcup_{j=1}^{k} \mathbf{Z}_{ij}$. Second, we should minimise the number of nodes that appear in all sets $\mathbf{Z}_i \in \mathcal{Z}$, and some of the candidate sets $\mathbf{Z}_i$ should be as small as possible. This maximises the chance that $\text{forb}(X, Y, \mathcal{G}_0) \cap \mathbf{Z}_i = \emptyset$ for at least one $\mathbf{Z}_i \in \mathcal{Z}$. Note that this is very different from the strategy proposed by Lu and White [2014].

**Example 2.** Consider the graphs from Figure 1 and the linear structural equation model compatible with $\mathcal{G}_0$, where all edge coefficients and error variances equal 1. We are interested in estimating the total effect $\tau_{yx}$, which here is simply the edge coefficient of the edge $X \to Y$ and therefore $\tau_{yx} = 1$ by the path tracing rules for total effects from Wright [1934].

We now illustrate for three candidate graphs that differ from $\mathcal{G}_0$, whether we can use tests for the null hypothesis $H_0$ to detect the mistakes in the candidate graphs and how this depends on the choice of sets $\mathcal{Z}$.

Consider the candidate graph $\mathcal{G}_1$ and the collection

$$\mathcal{Z} = \{\{A_1\}, \{A_1, A_2\}, \{A_1, A_2, R\}\}$$

of three valid adjustment sets relative to $(X, Y)$ in $\mathcal{G}_1$. A direct calculation shows that $\boldsymbol{\beta}_{yx.\mathcal{Z}} = (1.25, 1.25, 1.25)^\top$, none of which are equal to the total effect $\tau_{yx} = 1$. In this case, the null hypothesis $H_0$ is true even though the null hypothesis $H_0^*$ is false, i.e., testing $H_0$ will not detect that there is a mistake in the candidate graph. However, if we add the set of non-forbidden nodes $\mathbf{Z}_4 = \{A_1, A_2, B_1, B_2, V, D, R\}$ to $\mathcal{Z}$, then $H_0$ no longer holds as $\mathbf{Z}_4$ is a valid adjustment set in $\mathcal{G}_0$ and therefore $\beta_{yx.\mathbf{z}_4} = 1$. In this case testing $H_0$ will detect that there is a mistake in the candidate graph.

Consider now the candidate graph $\mathcal{G}_2$. It has exactly the same valid adjustment sets relative to $(X, Y)$ as the true graph $\mathcal{G}_0$. Testing $H_0$ will therefore not detect that there is a mistake in the candidate graph, irrespective of the collection of valid adjustment sets. Since the two graphs are equivalent with respect to estimating the total effect $\tau_{yx}$ with covariate adjustment this is not a concern.

Consider now the candidate graph $\mathcal{G}_3$. All valid adjustment sets relative to $(X, Y)$ in $\mathcal{G}_3$ result in estimates of $1.5$. Therefore testing $H_0$ will not detect that there is a mistake in the candidate graph, irrespective of the collection of valid adjustment sets. Interestingly, in the graph $\mathcal{G}_3'$ equal to $\mathcal{G}_3$ but with the edge $V \to Y$ removed, we can detect the mistakes by testing $H_0$ with, for example, the collection $\mathcal{Z}$ of all valid adjustment sets in $\mathcal{G}_3'$. This is an example of using an instrument to detect omitted variables bias [cf. Chen and Pearl, 2015], which our test implicitly exploits.

## 3.2 THE TEST STATISTIC

As a preparatory result and for completeness, we first derive the asymptotic distribution of the estimator $\hat{\boldsymbol{\beta}}_{yx.\mathcal{Z}}$.

**Lemma 3.** *Consider a p-dimensional random vector $\mathbf{V} = (V_1, V_2, \ldots, V_p)$ that follows a distribution where $\mathrm{E}(V_\ell^4) < \infty$ for all $1 \leq \ell \leq p$. Given two random variables $X, Y \in \mathbf{V}$, let $\mathcal{Z} = \{\mathbf{Z}_1, \mathbf{Z}_2, \ldots, \mathbf{Z}_k\}$, $k \geq 2$, be a collection of random subvectors of $\mathbf{V}$ that do not contain $X$ or $Y$, and let $\mathbf{Z}_i' = (X, \mathbf{Z}_i^\top)^\top$ for $i = 1, 2, \ldots, k$. Then the random vector $\sqrt{n}(\hat{\boldsymbol{\beta}}_{yx.\mathcal{Z}} - \boldsymbol{\beta}_{yx.\mathcal{Z}})$ converges in distribution to a multivariate normal random variable with mean zero and covariance matrix*

$$(\boldsymbol{\Sigma}_{\mathcal{Z}})_{ij} = \frac{\mathrm{E}(\delta_{x\mathbf{z}_i}\delta_{y\mathbf{z}_i'}\delta_{x\mathbf{z}_j}\delta_{y\mathbf{z}_j'})}{\mathrm{E}(\delta_{x\mathbf{z}_i}^2)\,\mathrm{E}(\delta_{x\mathbf{z}_j}^2)} \quad 1 \leq i, j \leq k, \quad (2)$$

*where $\delta_{y\mathbf{z}_i'} = Y - \boldsymbol{\beta}_{y\mathbf{z}_i'}\mathbf{Z}_i'$ and $\delta_{x\mathbf{z}_i} = X - \boldsymbol{\beta}_{x\mathbf{z}_i}\mathbf{Z}_i$.*

In general the covariance matrix $\boldsymbol{\Sigma}_{\mathcal{Z}}$ of the limiting normal distribution will not be of full rank, that is, the distribution will be degenerate. To illustrate this, we now give an example.

**Example 4.** Consider again the DAG $\mathcal{G}_0$ and the linear structural equation model from Example 2. Let

$$\mathcal{Z} = \{\{A_1, B_1\}, \{A_1, A_2, B_1\}, \{A_1, B_1, B_2\},$$
$$\{A_1, A_2, B_1, B_2\}\}.$$

The asymptotic covariance matrix $\boldsymbol{\Sigma}_{\mathcal{Z}}$ of $\hat{\boldsymbol{\beta}}_{yx.\mathcal{Z}}$ is the rank-3 matrix

$$\begin{pmatrix} 1.75 & 1.25 & 1.5 & 1 \\ 1.25 & 1.25 & 1 & 1 \\ 1.5 & 1 & 1.5 & 1 \\ 1 & 1 & 1 & 1 \end{pmatrix}.$$

We now reformulate and slightly generalise the null hypothesis $H_0$ from Section 3.1 as follows. Consider a pair of random variables $(X, Y)$ and a collection of random vectors $\mathcal{Z} = \{\mathbf{Z}_1, \ldots, \mathbf{Z}_k\}$. Define a contrast matrix $\boldsymbol{\Gamma} \in \mathbb{R}^{(k-1) \times k}$ such that $\boldsymbol{\Gamma}\mathbf{1} = \mathbf{0}$ and $\text{rank}(\boldsymbol{\Gamma}) = k - 1$ and consider the null hypothesis: $H_0 : \boldsymbol{\Gamma}\boldsymbol{\beta}_{yx.\mathcal{Z}} = \mathbf{0}$. Based on the joint asymptotic normality of $\hat{\boldsymbol{\beta}}_{yx.\mathcal{Z}}$ we now construct an asymptotically $\chi^2$-distributed test statistic for this null hypothesis.

**Definition 5** (Rank-$r$ Moore-Penrose inverse)**.** Consider the spectral decomposition of an $l \times l$ positive semidefinite matrix $\boldsymbol{\Delta} = \mathbf{P}\boldsymbol{\Lambda}\mathbf{P}^\top$, where $\mathbf{P}$ is the orthonormal matrix of eigenvectors and $\boldsymbol{\Lambda} = \text{diag}(\lambda_1, \lambda_2, \ldots, \lambda_l)$ with $\lambda_1 \geq \lambda_2 \geq \cdots \geq \lambda_l$ the ordered eigenvalues of $\boldsymbol{\Delta}$. The rank-$r$ Moore-Penrose inverse of $\boldsymbol{\Delta}$ is the matrix $\boldsymbol{\Delta}_r^\dagger = \mathbf{P}\boldsymbol{\Lambda}_r^\dagger\mathbf{P}^\top$, where $r \leq \text{rank}(\boldsymbol{\Delta})$ and $\boldsymbol{\Lambda}_r^\dagger = \text{diag}(1/\lambda_1, \ldots, 1/\lambda_r, 0, \ldots, 0)$.

**Theorem 6.** *Assume the same conditions as in Lemma 3. Let $\boldsymbol{\Sigma}_{\mathcal{Z}}$ be the covariance matrix of the limiting distribution of*

$\sqrt{n}(\hat{\boldsymbol{\beta}}_{yx.\mathcal{Z}} - \boldsymbol{\beta}_{yx.\mathcal{Z}})$ and given a $(k-1) \times k$ contrast matrix $\boldsymbol{\Gamma}$, define $\boldsymbol{\Delta}_{\mathcal{Z}} = \boldsymbol{\Gamma}\boldsymbol{\Sigma}_{\mathcal{Z}}\boldsymbol{\Gamma}^{\top}$. Suppose that $\hat{\boldsymbol{\Sigma}}_{\mathcal{Z}}$ is a consistent estimator of $\boldsymbol{\Sigma}_{\mathcal{Z}}$ and that $\hat{r}$ is a consistent estimator of $\operatorname{rank}(\boldsymbol{\Delta}_{\mathcal{Z}}) = r_0, 1 \leq r_0 \leq k-1$. Let $\hat{\boldsymbol{\Delta}}^{\dagger}_{\mathcal{Z},\hat{r}}$ denote the rank-$\hat{r}$ Moore-Penrose inverse of $\hat{\boldsymbol{\Delta}}_{\mathcal{Z}} = \boldsymbol{\Gamma}\hat{\boldsymbol{\Sigma}}_{\mathcal{Z}}\boldsymbol{\Gamma}^{\top}$. Then under the null hypothesis $H_0 : \boldsymbol{\Gamma}\boldsymbol{\beta}_{yx.\mathcal{Z}} = 0$, the test statistic

$$T_{\hat{r}}^2 = n(\boldsymbol{\Gamma}\hat{\boldsymbol{\beta}}_{yx.\mathcal{Z}})^{\top} \hat{\boldsymbol{\Delta}}^{\dagger}_{\mathcal{Z},\hat{r}} (\boldsymbol{\Gamma}\hat{\boldsymbol{\beta}}_{yx.\mathcal{Z}}) \quad (3)$$

converges in distribution to a $\chi^2_{r_0}$-distributed random variable.

We can estimate $\boldsymbol{\Sigma}_{\mathcal{Z}}$ consistently by plugging in sample residuals for the population level residuals in equation (2). We refer to this estimator as the plug-in estimator and denote it $\hat{\boldsymbol{\Sigma}}_{\mathcal{Z}}$. For a detailed argument, see Lemma 3 in the supplementary materials. Note also that for simplicity, we only consider the contrast matrix $\boldsymbol{\Gamma}$ with $1$ at the entries $(j, j)$ and $-1$ at the entries $(j, j+1)$ for $j = 1, 2, \ldots, k-1$, and with zeroes at the remaining entries in this paper.

## 3.3 THE DEGREES OF FREEDOM

To compute the Moore-Penrose inverse and the degrees of freedom for the test statistic in equation (3), it is necessary to know the rank of $\boldsymbol{\Delta}_{\mathcal{Z}}$. There are two possible approaches to this problem. The first is to estimate the rank $r_0$ with some estimate $\hat{r}$. The second approach is to select the candidate sets $\mathcal{Z}$ in a way that ensures the matrix $\boldsymbol{\Delta}_{\mathcal{Z}}$ is invertible. We now develop tools for both approaches.

### 3.3.1 Estimating the Degrees of Freedom

A standard approach to estimating the rank of a matrix from a noisy observation is information criterion based model selection. This is equivalent to conducting sequential hypothesis tests [Camba-Méndez and Kapetanios, 2009] for the possible ranks. In order to apply such model selection to the rank estimation of $\boldsymbol{\Delta}_Z$, we first derive that the half-vectorised plug-in estimator $\operatorname{vech}(\hat{\boldsymbol{\Delta}}_{\mathcal{Z}})$ based on the plug-in estimator $\hat{\boldsymbol{\Sigma}}_{\mathcal{Z}}$ is asymptotically normal.

**Proposition 7.** *Under the same conditions as in Lemma 3, $\sqrt{n}\operatorname{vech}(\hat{\boldsymbol{\Delta}}_{\mathcal{Z}} - \boldsymbol{\Delta}_{\mathcal{Z}}) \xrightarrow{d} \mathrm{N}(\mathbf{0}, \mathbf{C})$, where $\mathbf{C} = \boldsymbol{\Pi}\mathbf{F}\boldsymbol{\Pi}^{\top}$ for some positive semidefinite matrix $\mathbf{F}$, with scaling matrix $\boldsymbol{\Pi} = \mathbf{E}_l(\boldsymbol{\Gamma} \otimes \boldsymbol{\Gamma})\mathbf{D}_k$. Here, $\mathbf{E}_l$ is the $l(l+1)/2 \times l^2$ elimination matrix, $l = k - 1$ and $\mathbf{D}_k$ is the $k^2 \times k(k+1)/2$ duplication matrix.*

Based on Proposition 7 and a consistent estimator $\hat{\mathbf{C}}$ of the matrix $\mathbf{C}$, we may construct a rank estimation procedure from the minimum discrepancy function (MDF) test statistic [Cragg and Donald, 1997, Donald et al., 2007], which has

the form

$$n \min_{\operatorname{rank}(\tilde{\boldsymbol{\Delta}}_{\mathcal{Z}}) \leq r} \operatorname{vech}(\hat{\boldsymbol{\Delta}}_{\mathcal{Z}} - \tilde{\boldsymbol{\Delta}}_{\mathcal{Z}})^{\top} \hat{\mathbf{C}}^{-1} \operatorname{vech}(\hat{\boldsymbol{\Delta}}_{\mathcal{Z}} - \tilde{\boldsymbol{\Delta}}_{\mathcal{Z}}).$$
$$(4)$$

This procedure, however, has only been shown to be consistent if either $\mathbf{C}$ is of full rank [Cragg and Donald, 1997] or, in slightly adapted form, if the true rank of $\mathbf{C}$ is known [Ratsimalahelo, 2003].

As we cannot estimate the rank of $\mathbf{C}$ to estimate the rank of $\boldsymbol{\Delta}_{\mathcal{Z}}$, we instead propose using a simplified estimator based on the MDF statistic from equation (4) which we call the pseudo-MDF estimator:

$$\hat{r} = \operatorname*{argmin}_{r \in \{1, \ldots, k-1\}} \Big\{ n\| \operatorname{vech}(\hat{\boldsymbol{\Delta}}_{\mathcal{Z}} - \tilde{\boldsymbol{\Delta}}_{\mathcal{Z},r})\|_2^2 +$$
$$\log(n)r(k - 1 - (r-1)/2) \Big\}, \quad (5)$$

where $\tilde{\boldsymbol{\Delta}}_{\mathcal{Z},r}$ is the best rank-$r$ reconstruction of $\hat{\boldsymbol{\Delta}}_{\mathcal{Z}}$ based on spectral decomposition such that $\tilde{\boldsymbol{\Delta}}_{\mathcal{Z},r}\tilde{\boldsymbol{\Delta}}^{\dagger}_{\mathcal{Z},r} = \mathbf{I}$. Note that we effectively assume that the matrix $\hat{\mathbf{C}}^{-1}$ is the identity matrix. In doing so, we ignore the covariance structure between the elements of $\hat{\boldsymbol{\Delta}}_{\mathcal{Z}}$. Although the elements of $\hat{\boldsymbol{\Delta}}_{\mathcal{Z}}$ are likely correlated, the pseudo-MDF rank estimate nonetheless works well empirically (see Section 4).

### 3.3.2 Selecting $\mathcal{Z}$ to Ensure Full Rank

Depending on the choice of candidate sets $\mathcal{Z}$, the asymptotic covariance matrix $\boldsymbol{\Sigma}_{\mathcal{Z}}$ may be of full rank. This is, for example, trivially true if there is only one set in $\mathcal{Z}$. Whenever $\boldsymbol{\Sigma}_{\mathcal{Z}}$ is invertible, the matrix $\boldsymbol{\Delta}_{\mathcal{Z}}$ is also invertible. We now propose a strategy to select $\mathcal{Z}$, such that $\boldsymbol{\Sigma}_{\mathcal{Z}}$ is likely to be of full rank and that also follows the guidelines derived from Theorem 1 in Section 3.1.

**Lemma 8.** *Consider nodes $X$ and $Y$ in a DAG $\mathcal{G} = (\mathbf{V}, \mathbf{E})$ such that $Y \in \operatorname{de}(X, \mathcal{G})$. Consider a collection of*

$$\mathcal{Z} = \{\mathbf{Z}_1, \ldots, \mathbf{Z}_k\} \cup \{\operatorname{nonforb}(X, Y, \mathcal{G})\}$$

*where $\mathbf{Z}_i$, $i = 1, \ldots, k$, are minimal valid adjustment sets relative to $(X, Y)$ in $\mathcal{G}$. If $\mathbf{Z}_i \setminus (\cup_{j \neq i}\mathbf{Z}_j) \neq \emptyset$ for all $i = 1, \ldots, k$, $\operatorname{nonforb}(X, Y, \mathcal{G}) \setminus (\cup_i\mathbf{Z}_i) \not\perp_{\mathcal{G}} X$ and we sample the edge coefficients and error variances for a linear structural equation model compatible with $\mathcal{G}$ from a distribution $P$, such that $P$ is absolutely continuous with respect to the Lebesgue measure, then the asymptotic covariance matrix $\boldsymbol{\Sigma}_{\mathcal{Z}}$ for the random vector $\hat{\boldsymbol{\beta}}_{yx.\mathcal{Z}}$ is $P$-almost surely of full rank.*

In general, the collection of all minimal valid adjustment sets will not fulfil the distinct node condition of Lemma 8. It is, however, easy to prune the set of all minimal valid adjustment sets to obtain a subset that fulfils the conditions of Lemma 8 and still covers the same set of nodes as the collection of all minimal valid adjustment sets.

---
**Algorithm 1** Testing procedure
---
1: **Input**: Candidate graph $\mathcal{G}$, vertices $(X, Y)$, data $\mathcal{D}_n$, testing strategy $S \in \{\text{All}, \text{Min+}\}$

2: **Output**: $p$-value

3: **if** $S = \text{All}$ **then**

4:     Set $\mathcal{Z}$ as the collection of all valid adjustment sets relative to $(X, Y)$ in $\mathcal{G}$

5: **if** $S = \text{Min+}$ **then**

6:     Set $\mathcal{Z}$ as a pruned collection of all minimal valid adjustment sets relative to $(X, Y)$ in $\mathcal{G}$ plus the set of non-forbidden nodes

7: **for** each adjustment set $\mathbf{Z}$ in $\mathcal{Z}$ **do**

8:     Get sample regression residuals $\mathbf{r}_{xz}$ and $\mathbf{r}_{yz'}$ from data $\mathcal{D}_n$, where $\mathbf{Z}' = (X, \mathbf{Z})$

9: Compute $\hat{\mathbf{\Sigma}}_{\mathcal{Z}}$ and $\hat{\mathbf{\Delta}}_{\mathcal{Z}}$ with regression residuals

10: **if** $S = \text{All}$ **then**

11:     Estimate optimal rank $\hat{r}$ from $\hat{\mathbf{\Delta}}_{\mathcal{Z}}$ based on (5)

12: **if** $S = \text{Min+}$ **then**

13:     Set $\hat{r}$ as the cardinality of $\mathcal{Z}$ minus one

14: Compute test statistic

$$T_{\text{obs}}^2 = n(\mathbf{\Gamma}\hat{\boldsymbol{\beta}}_{yx})^\top \hat{\mathbf{\Delta}}_{\mathcal{Z},\hat{r}}^\dagger (\mathbf{\Gamma}\hat{\boldsymbol{\beta}}_{yx})$$

15: Calculate $p$-value $= 1 - F(T_{\text{obs}}^2)$ where $F(\cdot)$ is the cumulative distribution function of $\chi_{\hat{r}}^2$

---

## 3.4 THE TESTING PROCEDURE

We propose a testing procedure that, given a pair of nodes $(X, Y)$ in a candidate graph $\mathcal{G}$ and a data set, tests whether adjusting for the valid adjustment sets relative to $(X, Y)$ in $\mathcal{G}$ leads to estimators that converge to the same quantity. The procedure uses the test statistic from equation (3) and we propose two strategies to select the collection of valid adjustment sets $\mathcal{Z}$.

The first strategy, which we call All, considers all available valid adjustment sets. This strategy is likely to lead to the best power for the test, but it requires estimating the degrees of freedom for the test statistic's asymptotic distribution. This is a difficult problem (see Section 3.3.1) and as a result the solution we propose does not have a formal consistency guarantee, although it performs well in practice (see Section 4). In addition, computing all valid adjustment sets is very computationally expensive, especially for moderate to large graphs, and as a result this strategy may often not be computationally feasible.

The second strategy, which we call Min+, is to prune the collection of all minimal valid adjustment sets as explained in Section 3.3.2 and then add the set of all non-forbidden nodes, which under the assumption $Y \in \text{de}(X, \mathcal{G})$ is always a valid adjustment set (see Lemma 8 in the supplementary materials). This approach avoids estimating the degrees of

freedom but may lead to a loss of power. Note that if $Y \notin \text{de}(X, \mathcal{G})$ we would need to replace the non-forbidden nodes with another large set such as $\text{Adjust}(X, Y, \mathcal{G})$ [Perković et al., 2018]. However, if $Y \notin \text{de}(X, \mathcal{G})$ every set that d-separates $X$ and $Y$ is a valid adjustment set and therefore our problem reduces to checking d-separation statements, for which there exists a wide literature on conditional independence tests [e.g. Spirtes et al., 2000]. Therefore, we disregard this case.

Another major advantage of the Min+ strategy is that it avoids the computationally heavy task of computing all valid adjustment sets. The number of minimal valid adjustment sets is typically much smaller than the number of valid adjustment sets and as a result the polynomial-delay algorithm by Van der Zander et al. [2014], which we use to estimate the set of all minimal valid adjustment sets, is generally quite fast. We verify this in a small simulation study, where the Min+ strategy ran on sparse graphs of size up to 5000 (see Section C.3 in the Supplementary Materials).

We summarise the testing procedure in Algorithm 1. As discussed in Section 3.1, the test cannot detect all types of mistakes in $\mathcal{G}$, but it nonetheless serves as a simple and targeted robustness check.

**Example 9.** To illustrate our testing procedure, we revisit the linear structural equation model from Example 2 as well as the true graph and candidate graphs shown in Figure 1. In addition, we also consider the candidate graph $\mathcal{G}_3'$ which is the graph $\mathcal{G}_3$ with the edge $V \to Y$ deleted. To each candidate graph we apply the testing procedure with both testing strategies (see Algorithm 1). Recall that for the candidate graphs $\mathcal{G}_0$ and $\mathcal{G}_3$ the null hypothesis is true, while it is false for the candidate graphs $\mathcal{G}_1$ and $\mathcal{G}_3'$. We sample 100 data sets with $n = 25$ observations, 100 sets with $n = 100$ observations, and another 100 sets with $n = 400$ observations from the underlying linear structural equation model and apply our testing procedure to each of these data sets. The resulting $p$-values are shown as probability-probability plots against the standard uniform distribution in Figure 2. We explain the construction of these plots more thoroughly in Section C.1 of the Supplementary Materials. When the null hypothesis is true, we see that both strategies lead to close to uniform $p$-values, especially when $n > 25$. We do not consider $\mathcal{G}_2$ as it is equivalent to $\mathcal{G}_0$ in terms of valid adjustment sets. We also observe reasonable power for $n > 25$ and $\mathcal{G}_1$, especially with strategy All. For $\mathcal{G}_3'$ on the other hand the power is mediocre except for All and $n = 400$.

## 4 SIMULATIONS

We investigate the finite sample performance of the testing procedure from Algorithm 1 in a simulation study. The study is structured as follows. We randomly generate 50 DAGs for each graph size from $\{10, 15\}$. The expected

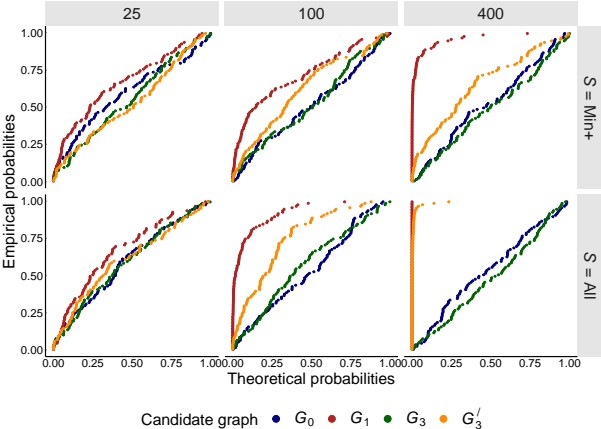

Candidate graph ● $G_0$ ● $G_1$ ● $G_3$ ● $G_3'$

Figure 2: Probability-probability plots of the $p$-values in Example 9. Theoretical probabilities are from the cumulative distribution function of a standard uniform distribution. Rows: test strategies. Columns: sample sizes for the test.

neighbourhood size is sampled uniformly from $\{2, 3, 4, 5\}$ for each graph. Then, for each DAG $\mathcal{G}_0$ we randomly generate a compatible linear structural equation model by (i) sampling the edge coefficients uniformly from the interval $[-2, -0.1] \cup [0, 1, 2]$, (ii) sampling the error distribution uniformly to either be normal, $t$, uniform or logistic for all errors and (iii) uniformly sampling scale parameters which depend on the error distribution such that the error variances are in the interval $[0.4, 1.6]$. We randomly choose a pair of nodes $(X, Y)$ such that $Y \in \mathrm{de}(X, \mathcal{G}_0)$ and that there exist at least two valid adjustment sets relative to $(X, Y)$ in $\mathcal{G}_0$.

For each true DAG $\mathcal{G}_0$ we then sample 40 data sets from the corresponding linear structural equation model. The sample size $m$ is 100 for half of these data sets, and 400 for the other half. With each of these data sets we estimate a causal graph $\mathcal{G}$ using either the Greedy Equivalence Search (GES) algorithm [Chickering, 2002], if the errors of the linear structural equation model are normal or the Linear Non-Gaussian Acyclic Models (LiNGAM) algorithm [Shimizu et al., 2006], otherwise. We do this to generate a large number of plausible candidate causal graphs for our testing procedure. We use the sample sizes 100 and 400 to ensure that some of the candidate graphs contain more errors and some fewer. We refer to the candidate graphs that were generated using the sample size 100 as low accuracy candidate graphs and to those that were generated with the sample size 400 as high accuracy ones.

For each candidate graph $\mathcal{G}$ and each sample size $n \in \{50, 100, 200, 400\}$ we do the following procedure. We sample an additional 100 data sets with sample size $n$ from the corresponding true linear structural equation model. Given these data sets, the pair $(X, Y)$ and the candidate graph $\mathcal{G}$, we apply Algorithm 1 using both strategies for graphs with 10 or 15 nodes. To measure the performance of

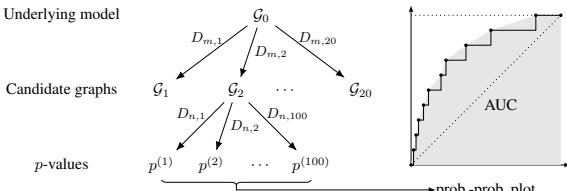

Figure 3: An illustration of the double simulation scheme for the simulation study and an illustration of the AUC metric.

our testing procedure we then compute the area under the curve (AUC) of the probability-probability plot of the 100 $p$-values against the uniform distribution. This means that in total we obtain 8 AUCs for each candidate graph, i.e., 4 for each testing strategy and 2 for each sample size. Figure 3 gives an illustration of the layered simulation scheme. We give further details for the design of the simulation study in Section C.2 of the Supplementary Materials.

Figure 4 is an ensemble of violin plots of the AUCs from the simulation study. As we have access to the true graph, we can decide for each candidate graph and testing strategy, whether the null hypothesis for the test, i.e. $H_0$, is true or false and plot these cases separately. Figure 4 shows that as the sample size of the data set used for our testing procedure increases, the AUCs in the cases where the null hypothesis is true become centred around 0.5. This indicates that both strategies control the type-I error rate asymptotically. There are, however, very large and small AUCs when we use the strategy $S = \mathrm{All}$ with small sample sizes and for the more accurate candidate graphs. This is likely due to the rank estimation step required for this strategy and indicates that as expected the strategy $\mathrm{Min}+$ is more stable than $\mathrm{All}$.

In the cases where the null hypothesis is false, i.e., the candidate graph contains a mistake that the test can detect, the AUCs have a cluster close to 1 which is especially pronounced for the larger sample sizes. This indicates that our testing procedure has good power in many cases. Unsurprisingly, the AUCs are smaller for the candidate graphs with fewer errors, since it is more difficult to detect that there is a mistake in an almost correct graph than in a glaringly incorrect one. Nonetheless, the AUCs remain respectable and there continue to be AUCs close to 1. In general, the AUCs for strategy $\mathrm{All}$ are larger than those for $\mathrm{Min}+$, although this gain is obtained at the price of a loss in stability.

The AUCs we consider do not capture the behaviour of our testing procedure fully, so we also calculate the proportion of tests rejected at level 0.05 among all tests performed in the simulation study as an additional metric. The results are given in Table 1. They indicate that for both strategies our testing procedure controls the type-I error rate asymptotically and at the same time has good power for the alternative.

Note that for conciseness we have only analysed the performance of our procedure for testing the test null $H_0$ and

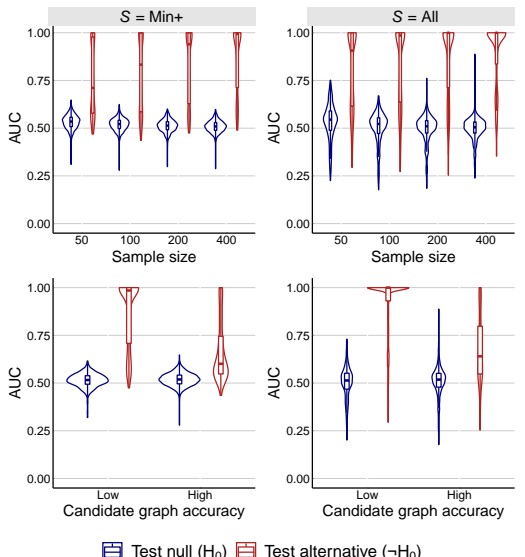

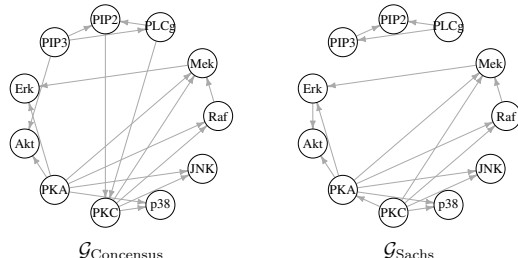

Figure 5: Causal DAGs representing intracellular signalling network among human primary naïve CD4$^+$ T cells.

Figure 4: Violin plots (layered with boxplots) of the areas under the curve (AUC) from the simulation study. The AUCs are grouped by sample size for the testing procedure (first row) and the expected accuracy of the candidate graph (second row).

| Cand. graph | | Null ($H_0$) | | Alternative ($\neg H_0$) | |
|---|---|---|---|---|---|
| accuracy | $n$ | $S = \text{Min+}$ | $S = \text{All}$ | $S = \text{Min+}$ | $S = \text{All}$ |
| Low | 50 | 0.0753 | 0.0903 | 0.5570 | 0.7396 |
| | 100 | 0.0634 | 0.0626 | 0.6352 | 0.7880 |
| | 200 | 0.0540 | 0.0510 | 0.7132 | 0.8341 |
| | 400 | 0.0494 | 0.0468 | 0.7887 | 0.8812 |
| High | 50 | 0.0781 | 0.0887 | 0.1543 | 0.1697 |
| | 100 | 0.0631 | 0.0585 | 0.2026 | 0.2094 |
| | 200 | 0.0559 | 0.0499 | 0.2838 | 0.3010 |
| | 400 | 0.0540 | 0.0471 | 0.3838 | 0.4152 |

Table 1: Proportion of hypotheses rejected at level $0.05$ in the simulation study.

not the stricter $H_0^*$. However, since $H_0$ is in fact the null-hypothesis our procedure is formally testing, the performance does not differ meaningfully between the two cases that (i) $H_0^*$ holds and that (ii) $H_0^*$ does not hold but $H_0$ does. We verify this in Section 4 of the Supplementary Materials. We also investigate how often the problematic case that $H_0$ holds but $H_0^*$ does not occur in our simulation study, i.e., the testing procedure has no power to detect a meaningful mistake in the candidate graph: it never occurs in more than $15\%$ of the cases where $H_0$ holds although the actual percentage is much lower for some settings of our simulation study (see Table 1 in the Supplementary Materials).

We also investigate the performance of our testing procedure in an additional simulation study with graphs of size 20, 40 and 80. Here, we only consider the $\text{Min+}$ strategy as the All is too computationally expensive. Due to space constraints we provide the results in Section C.3 of the Supplementary

Materials, but they do not differ meaningfully from the results for the smaller graphs.

# 5 REAL DATA EXAMPLE

We apply our testing procedure to the single cell data collected for the investigation of human primary naïve CD4$^+$ T cell signalling networks by Sachs et al. [2005]. This data set consists of measurements from a total of 9 experimental conditions. We will only use the data from the observational regime, which corresponds to the experimental setting with reagent anti-CD3/CD28. This subset of the data consists of $853$ measurements of $11$ phosphorylated proteins and phospholipids. The observational data is thought to be consistent with the conventionally accepted molecular interaction network (also called the consensus graph, $\mathcal{G}_{\text{Consensus}}$, Figure 5 left). We use alternative graph proposed in Sachs et al. [2005] ($\mathcal{G}_{\text{Sachs}}$, Figure 5 right) to evaluate the results of the analysis.

We consider $\mathcal{G}_{\text{Consensus}}$ as the candidate graph $\mathcal{G}$. We extract all pairs of nodes $(X, Y)$ that satisfy $Y \in \text{de}(X, \mathcal{G})$. There are 36 such node pairs in the graph. For every pair, we apply our testing procedure with strategy All to the log-transformed and centred observational data. After a Bonferroni correction only the $p$-values for the pairs (PKA, Erk) and (PKA, Akt) are significant at $0.05$ level ($1.19 \times 10^{-14}$ and $4.91 \times 10^{-14}$).

We now take a closer look at these two node pairs. The collection $\mathcal{Z}$ of all valid adjustment sets relative to (PKA, Erk) in the consensus graph consists of 419 sets. The test rejects the null hypothesis that these adjustment sets lead to estimates of the same quantity. To illustrate a potential error in $\mathcal{G}_{\text{Consensus}}$, we consider the valid adjustment sets $\emptyset$ and {PLCg, PIP2, PIP3, Akt, PKC, p38, JNK}. If we consider the alternative graph $\mathcal{G}_{\text{Sachs}}$ as a more appropriate representation of the true data generating mechanism, the rejection is justifiable. The covariate Akt is a forbidden node in $\mathcal{G}_{\text{Sachs}}$ because it opens a collider path PKA $\rightarrow$ Akt $\leftarrow$ Erk. On the other hand, the empty set is a valid adjustment set also in $\mathcal{G}_{\text{Sachs}}$. A similar argument applies to the the pair

(PKA, Akt). The collection $\mathcal{Z}$ relative to this node pair also has a size of $419$, among which we can look at adjustment sets $\emptyset$ and {Raf, Mek, PLCg, PIP2, PIP3, Erk, PKC, p38, JNK}. Using Erk is problematic as it is a forbidden node in $\mathcal{G}_{\text{Sachs}}$ because it blocks the causal path PKA $\rightarrow$ Erk $\rightarrow$ Akt. This indicates that in both cases our testing procedure is detecting a mistake in the consensus graph.

Our testing procedure produces rank estimates of $\boldsymbol{\Delta}_{\mathcal{Z}}$ mostly at $1$ ($30$ out of $36$ cases), even though the size of $\mathcal{Z}$ goes up to $419$. This illustrates how a large number of adjustment sets does not necessarily mean a large number of effective over-identifying constraints on the total effect for the test. It is unsurprising that our testing procedure with the Min+ strategy detects the same two pairs of nodes as problematic (p-values $7.04 \times 10^{-15}$ and $7.35 \times 10^{-15}$).

# 6   CONCLUSION AND DISCUSSION

In this paper, we propose a robustness test that checks whether it is reasonable to use a candidate causal graph to estimate a total effect of interest with covariate adjustment. This is a useful model validation tool for practitioners who wish to estimate a total effect with covariate adjustment and rely on causal graphs obtained from domain knowledge.

We develop our testing procedure assuming that the candidate graph is a DAG. In applications with unmeasured confounding between the covariates, it is more natural to assume that the candidate graph is an acyclic directed mixed graph (ADMG) with bi-directed edges that represent error correlations induced by the presence of unmeasured confounding. If the candidate ADMG contains at least two valid adjustment sets, it is also possible to apply our testing procedure in this setting with one limitation. The set $\text{nonforb}(X, Y, \mathcal{G})$ may not be a valid adjustment set and as a result the strategy Min+ fails. We believe it is possible to adapt Min+ to an ADMG by replacing $\text{nonforb}(X, Y, \mathcal{G})$ with a suitable alternative large valid adjustment set but we leave this for future research.

Another interesting idea for future research is that in general, given a valid adjustment set and a forbidden node, adding the node to the set should change the limit of the resulting estimator. It may be possible to exploit this in order to devise a testing procedure similar to the one proposed in this paper but that also exploits the information contained in the forbidden nodes of the candidate causal graph.

### Acknowledgements

We thank Milan Kuzmanovic for proposing the idea for Lemma 8. We also thank Vi Thanh Pham, Nicola Gnecco and Jonas Peters for feedback and insightful discussions. LH was supported by a research grant (18968) from VILLUM FONDEN.

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
