# OpenReview forum: "A Robustness Test for Estimating Total Effects with Covariate Adjustment"
_auai.org/UAI/2022/Conference — UAI 2022 Poster_

### Official Review · Reviewer_QSkx · 2022-04-08

**Q2(1) Originality/Novelty:** 3
**Q2(2) Significance/Impact:** 2
**Q2(3) Correctness/Technical Quality:** 2
**Q2(6) Clarity Of Writing:** 3
**Q6 Overall Score:** 4
**Q8 Confidence In Your Score:** 3

**Q1 Summary And Contributions:**

The authors propose a test that when rejected indicates that the causal graph at hand may be incorrect. The test builds on the fact that adjusting for different valid sets should result in the say estimate of the causal effect. The asymptotic null distribution of the test is derived. The test is evaluated experimentally with simulated and real data.

**Q2 Assessment Of The Paper:**

More detailed information regarding each of these aspects is given below:

**Q2(4) Quality Of Experiments (Optional):**

2: Fair: The experimental evaluation is weak: important baselines are missing, or the results do not adequately support the main claims.

**Q2(5) Reproducibility:**

3: Good: Key resources (e.g., proofs, code, data) are available and key details (e.g., proofs, experimental setup) are sufficiently well-described for competent researchers to confidently reproduce the main results.

**Q3 Main Strengths:**

The authors propose a test that when rejected indicates that the causal graph at hand may be incorrect. The test builds on the fact that adjusting for different valid sets should result in the say estimate of the causal effect. The asymptotic null distribution of the test is derived. Imho this is the main strength of the paper, together with the two approaches to select the valid adjusting tests in the test.

**Q4 Main Weakness:**

The paper has some passages that need clarification. In page 4, the authors claim that controling the type I error of H0 controls the type I error of H0*. I am sorry but I do not quite get it. I do not get either the relevance of Theorem 1 for building the test, since it involves a non valid adjustment set Z_i.

Another main weakness is that wrong causal graphs can go undetected, as illustrated with G3 in page 4. The experiments show that the test controls the type I error for finite samples. Is this the type I error of H0 or H0*? I am inclined to believe it is H0, which is not what we want to control.

**Q5 Detailed Comments To The Authors:**

The paper has some passages that need clarification. In page 4, the authors claim that controling the type I error of H0 controls the type I error of H0*. I am sorry but I do not quite get it, as H0* will be rejected every time that H0 is rejected and some times when H0 is not rejected, i.e. when all the betas are equal but unequal tau. Please, clarify this. I do not get either the relevance of Theorem 1 for building the test, since it involves a non valid adjustment set Z_i, whereas the test only involves valid adjustment sets. For the same reasons, I do not get either the paragraph under the theorem, i.e. the guidelines to construct the adjustment sets in the test. How do they follow from the theorem?

Another main weakness is that wrong causal graphs can go undetected, as illustrated with G3 in page 4. That is, the power of the test may be limited. The authors conclude through experiments that the power is good. I guess this is open to discussion. The experiments show also that the test controls the type I error for finite samples. Is this the type I error of H0 or H0*? I am inclined to believe it is H0, which is not what we want to control. Please clarify this. The experiments with real data are nice but I am not convinced about the use of G_Sachs. If I got it right, the authors use G_Sachs to justify the rejection of G_Consensus. However, without further explanation, I find it harder to believe that Sachs et al. are right and the "the conventionally accepted molecular interaction G_Consensus" is wrong.

**Q7 Justification For Your Score:**

There are key paragraphs that need further clarification before I can raise my score. I am uncertain whether the power of the test is sufficiently good.

**Q9 Complying With Reviewing Instructions:**

1: Yes.

---

### Official Review · Reviewer_u79W · 2022-04-09

**Q2(1) Originality/Novelty:** 3
**Q2(2) Significance/Impact:** 3
**Q2(3) Correctness/Technical Quality:** 3
**Q2(6) Clarity Of Writing:** 3
**Q6 Overall Score:** 6
**Q8 Confidence In Your Score:** 4

**Q1 Summary And Contributions:**

The paper proposes a robustness test for the task of estimating the total causal effect when there are multiple valid adjustment sets in the candidate causal graph. If the test rejects,  then it is not reasonable to use this candidate causal graph for estimating a total causal effect with covariate adjustment. The proposed method relies on the causal graph given from domain knowledge.

**Q2 Assessment Of The Paper:**

More detailed information regarding each of these aspects is given below:

**Q2(4) Quality Of Experiments (Optional):**

3: Good: The experimental evaluation is adequate, and the results convincingly support the main claims.

**Q2(5) Reproducibility:**

4: Excellent: Key resources (e.g., proofs, code, data) are available and key details (e.g., proof sketches, experimental setup) are comprehensively described for competent researchers to confidently and easily reproduce the main results.

**Q3 Main Strengths:**

The method proposed in this work was interesting and useful for one who wants to estimate a total causal effect by covariate adjustment and has the candidate causal graph.
The test is extended from the robust test in Lu and White [2014] by using a causal graph to determine $\mathbf{z}$ and $forbind()$
The experiments also show the proposed test is able to detect a mistake in the consensus graph.

**Q4 Main Weakness:**

1. The test is a necessary condition if my understanding is correct. One can not draw any conclusions from some causal graphs.
2. The time complexity of the testing procedure may be too high so it is inappropriate for complex causal graphs.

**Q5 Detailed Comments To The Authors:**

*Regard ''Example 2'', how did we know $\tau_{yx} =1$? In real-world applications, it is impossible to check the validity of Eq.(1) since we did not have the ground truth of the total causal effect. Moreover, the example is not clear on how the testing procedure works.

*It is easy to follow that $Z$ should cover as many nodes in the candidate graph G as possible, but for the second, how do we get Z as small as possible? Let's continue to see the example 2, in $Z$ = {{$A_1$}, {$A_1, A_2$}, {$A_1, A_2, R$}},  {$A_1, A_2, R$} is not the minimal adjustment set in $\mathcal{G}_1$ since R is redundant.

*In Line 8 of Algorithm 1,  what is $\mathbf{r}_{y\mathbf{z}'}$?

**Q7 Justification For Your Score:**

The quality of the candidate causal graph is a crucial factor for the accuracy of the testing procedure.
Another concern is the time complexity, which seems very high and the applications of the testing procedure will be limited.

**Q9 Complying With Reviewing Instructions:**

1: Yes.

---

### Official Review · Reviewer_tvTL · 2022-04-13

**Q2(1) Originality/Novelty:** 3
**Q2(2) Significance/Impact:** 3
**Q2(3) Correctness/Technical Quality:** 3
**Q2(6) Clarity Of Writing:** 4
**Q6 Overall Score:** 7
**Q8 Confidence In Your Score:** 4

**Q1 Summary And Contributions:**

The paper proposes tests to protect against misspecification of a causal graph when performing effect estimation via backdoor adjustment in settings where there are no unmeasured confounders. The ideas presented are novel and interesting. While the method is limited to linear structural equation models, it has the potential to be extended to settings with unmeasured confounders.

**Q2 Assessment Of The Paper:**

More detailed information regarding each of these aspects is given below:

**Q2(4) Quality Of Experiments (Optional):**

4: Excellent: The experimental evaluation is comprehensive and the results are compelling.

**Q2(5) Reproducibility:**

4: Excellent: Key resources (e.g., proofs, code, data) are available and key details (e.g., proof sketches, experimental setup) are comprehensively described for competent researchers to confidently and easily reproduce the main results.

**Q3 Main Strengths:**

- Tackles an important problem of performing robustness checks against misspecification of a causal graph for downstream estimation tasks
- The paper is clearly written: the motivation for the algorithm, assumptions for the method, and the algorithm itself are clearly presented
- The experiments do a good job backing the theory

**Q4 Main Weakness:**

1. The paper does a good job overall connecting to related work. There's just one thread of work that I think is closely related and may have been overlooked: these works involve performing independence tests of the form A \indep Y |  X, Z where A is an "anchor" variable, Y is the outcome, X is the treatment, and Z is the proposed conditioning set. Under certain assumptions related to the anchor, if this test succeeds, Z passes the check for being a valid adjustment set without requiring knowledge of the structure of the whole graph. These methods also happen to be non-parametric (at least in principle.) An empirical comparison to these works is not necessary, but a short discussion comparing the two theories would be interesting (links to papers are in the detailed comments below.) Though not directly related, there is also similar work on this for checks for validity of front-door adjustment sets.

**Q5 Detailed Comments To The Authors:**

Re: 1 in Q4, here are some papers related to robustness checks with an anchor variable for backdoor (there may be a couple more useful references for related papers that you can find in the related work of the latter paper):
- http://proceedings.mlr.press/v31/entner13a.pdf
- https://arxiv.org/pdf/2106.11560.pdf

Other comments -- these are mostly suggestions for things that may be interesting to discuss:

- Given examples where it seems like even if we consider all valid adjustment sets it may be impossible to detect inaccuracies in the posited graph (and where it is not harmless in that it also leads to biased estimates), is there ever a time where it may be beneficial to incorporate nodes that would be considered forbidden in the graph we've drawn given that we have some suspicion that it may be wrong in the first place?

- I found the concluding discussion regarding ADMGs quite interesting, thank you for that. Sort of related -- are there some ideas from the current proposal that also extend to non-parametric settings?

**Q7 Justification For Your Score:**

A solid paper with interesting ideas. The linearity assumption limits the applicability a little, but the present work opens up some interesting avenues of research into robustness checks for causal estimation procedures in different settings -- with unmeasured confounders for example.

**Q9 Complying With Reviewing Instructions:**

1: Yes.

---

### Official Review · Reviewer_TYEp · 2022-04-14

**Q2(1) Originality/Novelty:** 3
**Q2(2) Significance/Impact:** 2
**Q2(3) Correctness/Technical Quality:** 3
**Q2(6) Clarity Of Writing:** 2
**Q6 Overall Score:** 7
**Q8 Confidence In Your Score:** 3

**Q1 Summary And Contributions:**


The authors investigate the problem whether a causal DAG correctly represents the underlying data set, especially the causal effect of a node X on Y in the data.
To do so, they estimate the causal effect with different adjustment sets.
If the graph is correct, all these estimations must return the same value.
They perform statistical analyses and experiments to see how likely it is that .


**Q2 Assessment Of The Paper:**

More detailed information regarding each of these aspects is given below:

**Q2(4) Quality Of Experiments (Optional):**

3: Good: The experimental evaluation is adequate, and the results convincingly support the main claims.

**Q2(5) Reproducibility:**

3: Good: Key resources (e.g., proofs, code, data) are available and key details (e.g., proofs, experimental setup) are sufficiently well-described for competent researchers to confidently reproduce the main results.

**Q3 Main Strengths:**

Verifying that one has the correct graph is an important task to solve when using graphical models.

With their algorithm they can often detect an invalid graph.

They have proofs and experiments.

**Q4 Main Weakness:**


Some parts are hard to understand and perhaps not explained well.

They seem to have missed a case in the proof of theorem 1

Their approach only works for linear models.

**Q5 Detailed Comments To The Authors:**


>Minimal valid adjustment sets

There has been much research on (finding) minimal and minimum valid adjustment sets you could have cited:

Johannes Textor and Maciej Liskiewicz. "Adjustment criteria in causal diagrams: An algorithmic perspective" 2011

van der Zander et al. "Constructing separators and adjustment sets in ancestral graphs" 2014

van der Zander et al. "Separators and adjustment sets in causal graphs: Complete criteria and an algorithmic framework" 2019

>Theorem 1:

What does "P-a.s." mean?

>p11 Clearly, p must contain at least one non-collider.

What if p is X -> Z <- Y? There is no non-collider, no node in forb, and yet p is open given Z.

>p11 By rewriting the ordinary least squares coefficients in this equation as functions of the covariances from the underlying linear causal model and multiplying out the denominators we can reformulate the equation βyx.zi − βyx.zj as a polynomial in the covariances.

How do you do this rewriting and reformulating? I am not sure I can follow here. Perhaps give some examples.

>p4 section 3.2

That is difficult math. I have trouble following it.

>p6 However, if Y ∈  / de(X, G) every set that d-separates X and Y is a valid adjustment set and therefore our problem reduces to checking d-separation statements,  for which there exists a wide literature on conditional in-dependence tests [e.g. Spirtes et al., 2000]. Therefore, we disregard this case.

Every set without the forbidden nodes that d-separates X and Y in the proper back-door graph is also a valid adjustment set, so can you not reuse that literature for either case?

>p7 Unsurprisingly, the AUCs are smaller for the candidate graphs with fewer errors, although they remain respectable and there continue to be AUCs close to 1

Shouldn't fewer errors lead to better results and thus higher AUCs?

>Figure 2

What are these probabilities?

>p8 After a Bonferroni adjustment

Perhaps say Bonferroni correction, since it is not an adjustment set

> Table 1:

Does column "null" means the null hypothesis is true (i.e. the graph is correct), and "alternative" the graph is not correct?
What are the hypotheses, the number of graphs or the number of adjustment sets?
What is a low and high accuracy?


There is some weird grammar:

>p2 "Second, we do not lose power completely against mistakes in the graph we had power for when using all valid adjustment sets."
>p3 "It is in this sense that the testing procedure is targeted."
>p4 We can can



**Q7 Justification For Your Score:**

There is a balance between strengths and weaknesses

update: raised technical and final score, because the path X->Z<-Y argument makes sense

**Q9 Complying With Reviewing Instructions:**

1: Yes.

---

### Decision · Program_Chairs · 2022-05-15

**Decision:**

Accept (Poster)

**Comment:**

Meta Review: This paper proposes a test of a causal DAG that is based on comparing estimation results from different adjustment sets, which according to the DAG should all coincide. If evidence is found for substantial heterogeneity between these estimates, this would point to possible errors in the initial DAG. The method is similar to coefficient stability tests used in econometrics.

Pros:

Reviewers found the problem of detecting errors in DAGs to be important. They felt that the paper does a good job in combining theoretical arguments (such as deriving the null distribution) and empirical data to support its claims.

Cons:

Reviewers pointed out that some errors in DAGs cannot be detected using this method. (For example, some modifications to the input DAG do not change the valid adjustment sets and will therefore not be found.) Also, several reviewers found that important aspects of the paper were not clearly explained. Unfortunately, most reviewers did not participate in the discussion, so that it is not clear whether these clarity issues were resolved to their satisfaction by the authors' detailed reply. However, the authors did give specific suggestions for addressing these concerns in their replies, and the reviewer who did participate in the discussion raised their score and was satisfied with the answer. Overall I therefore argue that this paper should be accepted.